# Genetic Diversity and Population Structure of *Alnus cremastogyne* as Revealed by Microsatellite Markers

**Hong-Ying Guo [1,2,3,4], Ze-Liang Wang [2,4], Zhen Huang [1,2,3,4], Zhi Chen [4], Han-Bo Yang [4] and Xiang-Yang Kang [1,2,3,4],***

[1] Beijing Advanced Innovation Center for Tree Breeding by Molecular Design, Beijing Forestry University, Beijing 100083, China; hyguosaf2019@hotmail.com (H.-Y.G.); wlhz1314@gmail.com (Z.H.)

[2] National Engineering Laboratory for Tree Breeding, Beijing Forestry University, Beijing 100083, China; wzl090716@gmail.com

[3] Beijing Laboratory of Urban and Rural Ecological Environment, Beijing Forestry University, Beijing 100083, China

[4] Sichuan Academy of Forestry, Chengdu 610081, Sichuan, China; chen610081@126.com (Z.C.); yanghanbo6@163.com (H.-B.Y.)

**\*** Correspondence: kangxy@bjfu.edu.cn; Tel.: +86-10-62336168

**Abstract:** *Alnus cremastogyne* Burk. is a nonleguminous, nitrogen-fixing tree species. It is also the most important endemic species of *Alnus* Mill. in China, possessing important ecological functions. This study investigated population genetic variation in *A. cremastogyne* using 175 trees sampled from 14 populations native to Sichuan Province with 25 simple sequence repeat (SSR) markers. Our analysis showed that *A. cremastogyne* has an average of 5.83 alleles, 3.37 effective alleles, an expected heterozygosity of 0.63, and an observed heterozygosity of 0.739, indicating a relatively high level of genetic diversity. The *A. cremastogyne* populations in Liangshan Prefecture (Meigu, Mianning) showed the highest level of genetic diversity, whereas the Yanting population had the lowest. Our analysis also showed that the average genetic differentiation of 14 *A. cremastogyne* populations was 0.021. Analysis of molecular variance (AMOVA) revealed that 97% of the variation existed within populations; only 3% was among populations. Unweighted pair-group method with arithmetic means (UPGMA) clustering and genetic structure analysis showed that the 14 *A. cremastogyne* populations could be clearly divided into three clusters: Liangshan Prefecture population, Ganzi Prefecture population, the other population in the mountain area around the Sichuan Basin and central Sichuan hill area, indicating some geographical distribution. Further analysis using the Mantel test showed that this geographical distribution was significantly correlated with elevation.

**Keywords:** *Alnus cremastogyne*; SSR; genetic diversity; population structure; genetic improvement; polyploid

## 1. Introduction

The genus *Alnus* Mill. (commonly known as alder) is a nonleguminous, nitrogen-fixing tree species, with a root system rich in nodules. These can improve soil, are important pioneer tree species for forestation, and have important ecological functions. Alder is the most ancient genus in the Betulaceae family; it is also an important plant group in the Cenozoic flora in the northern hemisphere. Alder is mostly distributed in Eurasia and North America, and is rarely seen in Latin America and Africa, with 11 species in China [1,2]. Sichuan Province and the adjacent areas are important alder distribution regions where *A. cremastogyne*, *A. ferdinandi-coburgii*, *A. lanata*, and *A. nepalensia* grow, and may possibly be the site of origin and differentiation for alder [1]. Among these four species,

*A. cremastogyne* is the most important endemic Chinese species and is the most widely studied. *A. cremastogyne* is highly adaptable, has a short juvenile period, high seed yield, and rapid growth rate. Currently, its suitable planting area has been expanded to the middle and lower areas of the Yangtze River. *A. cremastogyne* has become an important tree species in the Returning Farmland to Forest Program, ecological reconstruction programs, and mixed-species forest plantation in the Yangtze River valley in China.

Genetic improvement is largely based on a detailed understanding of genetic variation. Current investigations on the genetic improvement of *A. cremastogyne* mainly focus on conventional breeding. Studies on population genetic variation of *A. cremastogyne* mostly use approaches for phenotyping [3–6]. There are very few reports utilizing molecular markers, with only one study by Zhuo et al. [7] who used random amplified polymorphic DNA (RAPD) markers to study genetic variation in 12 *A. cremastogyne* provenance populations [7]. Rao et al. [8] is the only study that developed simple sequence repeat (SSR) markers for alder using transcriptome data from three species: *A. cremastogyne*, *A. glutinosa*, and *A. firma* [8]. Briefly, population genetic variation in *A. cremastogyne* lacks supporting data at the molecular level, which in turn limits the species' protection and utilization.

Simple sequence repeat (SSR) molecular markers occur abundantly in genomes, and many are highly polymorphic, stable, and exhibit codominance, and thus these have been extensively used to dissect plant genetic diversity, population structure, and evolutionary history. The earliest report by Zhuk et al. found that eight of 15 birch SSR primer pairs yielded amplification products in in *A. glutinosa* and *A. incana* [9]. Subsequently, Lance et al. [10] designed 19 primer pairs from an enriched library of *A. maritima* and used them to amplify *A. maritima* and *A. serrulata*. Using SSR primers developed by Lance et al., Jones et al. systemically analyzed the genetic diversity, population structure, and mating system of the endangered species *A. maritima* in the United States, providing a theoretical foundation for conservation biology [11,12]. Subsequently, SSR technologies were applied to other alder species such as *A. glutinosa* and *A. incana*, and the population genetic diversity, genetic structure, evolution and phylogeny were gradually elucidated [13–17].

The present study investigated the genetic diversity, genetic structure, and variation patterns of *A. cremastogyne* populations within their natural distribution areas using SSR molecular markers. This study aims to facilitate the genetic improvement and utilization of *A. cremastogyne*.

## 2. Materials and Methods

### 2.1. Population Sampling

*Alnus cremastogyne* is naturally distributed in Sichuan Province and the adjacent areas, including Chengdu campagna, a mountainous area around the Sichuan Basin; the central Sichuan hill area, an alpine valley region of western Sichuan; and the mountainous area of southwest Sichuan. Trees in the study areas were typical natural secondary forests (Figure 1, Table 1), from which we sampled 175 individuals representing 14 populations. The sampled trees were separated by at least 50 m. Leaf samples were collected, dried with silica gel, transported back to the laboratory, and stored at $-70\ ^{\circ}$C until use.

### 2.2. DNA Extraction and Amplification

Genomic DNA was extracted from alder leaves using a genomic DNA extraction kit (Tiangen, Beijing, China). A total of 25 primer pairs (Table 2) for highly polymorphic loci were selected from three sources: published papers for *Alnus* species [9,10,18,19], the GenBank EST database of *A. glutinosa* [20], and our unpublished transcriptome data of *A. cremastogyne* [20]. The primers used in this study are shown in Table 2, and the forward primer was fluorescently labeled with 6-FAM.

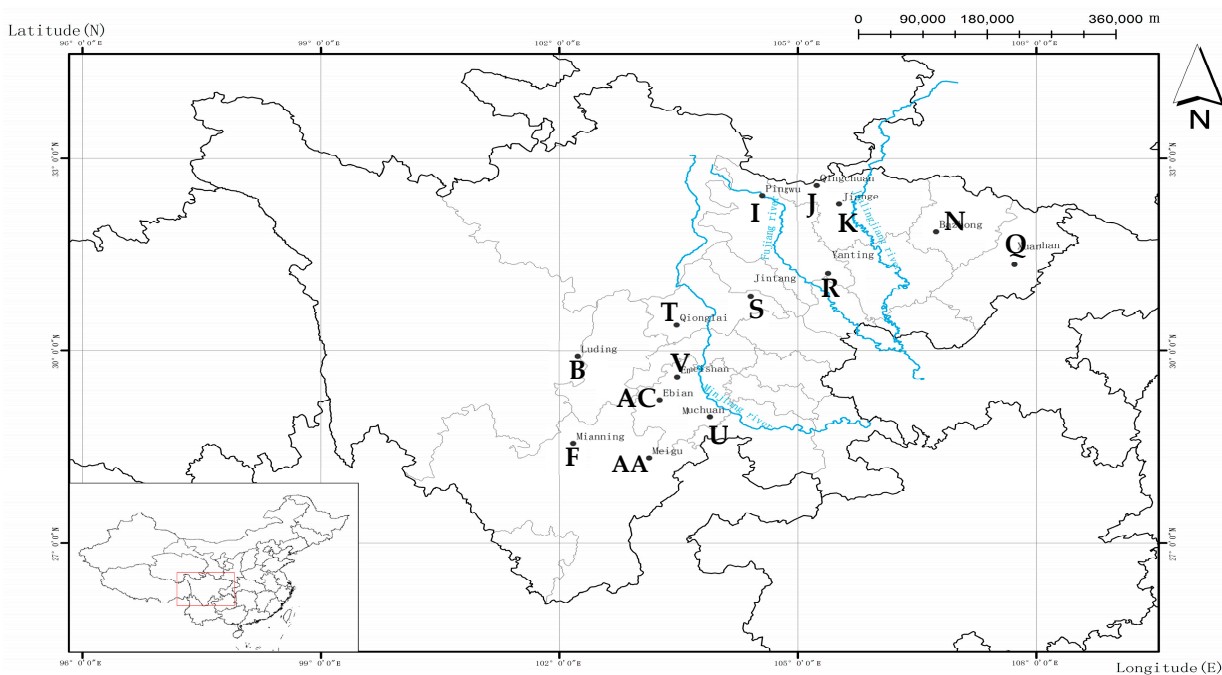

**Figure 1.** Locations in Sichuan Province of the sampled 14 populations of *Alnus cremastogyne.*

**Table 1.** Location and number of trees sampled in 14 *Alnus cremastogyne* populations in Sichuan Province.

| ID | Location | Sample Size | Longitude (E) | Latitude (N) | Elevation (m) |
|----|----------|-------------|---------------|--------------|----------------|
| AA | Meigu | 10 | 103.08 | 28.30 | 1609–2352 |
| AC | Ebian | 13 | 103.25 | 29.28 | 828–1180 |
| B | Luding | 14 | 102.13 | 29.77 | 1303–2343 |
| F | Mianning | 16 | 102.10 | 29.18 | 1884–1908 |
| I | Pingwu | 11 | 104.50 | 32.30 | 845–1440 |
| J | Qingchuan | 12 | 105.14 | 32.41 | 552–1496 |
| K | Jiange | 13 | 105.45 | 32.16 | 569–808 |
| N | Bazhong | 24 | 106.63 | 31.68 | 368–591 |
| Q | Xuanhan | 10 | 107.62 | 31.55 | 482–998 |
| R | Yanting | 7 | 105.30 | 30.89 | 380–530 |
| S | Jintang | 12 | 103.80 | 30.74 | 740–960 |
| T | Qionglai | 9 | 103.16 | 30.32 | 746–1118 |
| U | Muchuan | 16 | 103.71 | 29.15 | 500–530 |
| V | Emeishan | 8 | 103.33 | 29.59 | 650–1273 |

PCR was performed with Takara Taq (Takara, Dalian, China) in a final reaction volume of 10 μL, consisting of 5.92 μL of ddH$_2$O, 1.0 μL 10 × buffer, 1.0 μL 2.0 mM dNTP, 0.5 μL of each primer (at 5 μM), 1 μL DNA template, and 0.4 U Taq DNA polymerase. Amplification was conducted using the following cycling parameters: initial denaturation (5 min, 94 °C), followed by 25–30 cycles of denaturation (20 s, 94 °C), annealing (20 s, primer-specific annealing temperature), and extension (40 s, 72 °C), and a final extension for 10 min at 72 °C. Then, the PCR products were submitted for capillary electrophoresis on an ABI 3730xl DNA Analyzer (Applied Biosystems, Foster City, CA, USA), and the fragment lengths were determined using GeneMarker version 2.2.0 (SoftGenetics, State college, PA, USA).

**Table 2.** Characterization of 25 simple sequence repeat loci in *A. cremastogyne* based on 175 trees representing 14 populations in Sichuan Province.

| Locus | Repeat Motif | Primer Sequence (5′-3′) | Size Range | Tm (°C) | Na | Ne | Ho | Hs | H′t | G′st(Nei) | Nm | Primer Source |
|---|---|---|---|---|---|---|---|---|---|---|---|---|
| Acg3 | (CT)3CC(CT)2CC (CT)13AT(CT)5 | F: CTCCTTAGCTGGCACGGAC R: CCCTTCTTCATAAAACCCTCAA | 210–262 | 55 | 18 | 4.977 | 0.960 | 0.822 | 0.853 | 0.037 | 6.507 | L3.1 [9] |
| Acg7 | (AG)11 | F: CTGGGTTCAAACTTCCTTTGCT R: ACCCTAGGCCATGTTTGGTT | 265–293 | 56 | 13 | 3.770 | 0.949 | 0.761 | 0.779 | 0.024 | 10.167 | Alma7 [10] |
| Acg8 | (AG)12 | F: TGGCGACTATTAGAAGGACGA R: TGTGGGATAGCAAGTGTTGGA | 177–217 | 56 | 16 | 4.054 | 0.799 | 0.788 | 0.805 | 0.020 | 12.25 | Alma12 [10] |
| Acg11 | (GT)12(GA)10 | F: TGGGCTAGCATTAAGCACCA R: AAGGCCTCCCTTCCAACTTT | 248–256 | 58 | 5 | 2.244 | 0.891 | 0.580 | 0.577 | −0.006 | 1000 | Alma20 [10] |
| Acg16 | (AG)14 | F: GCAGACCAGAGTCTGTTATTCA R: AGACAATTTCGTGACTGGGTAT | 156–168 | 60 | 7 | 4.652 | 0.844 | 0.818 | 0.834 | 0.019 | 12.908 | CAC-A105 [18] |
| Acg19 | (TC)15 | F: CAGTCTATCTGCTACAAGCGTGGT R:GACGTTTTCAACGACCAAAAACAC | 130–138 | 58 | 5 | 2.674 | 0.902 | 0.652 | 0.669 | 0.025 | 9.75 | CAC-B113 [18] |
| Acg20 | (GA)12 | F: AAGCAAAATCCCAAGGTATCCAGT R: GGGGTTCCAACCAATTTATTCTTC | 142–196 | 58 | 25 | 2.464 | 0.660 | 0.621 | 0.631 | 0.016 | 15.375 | CAC-C118 [18] |
| Acg21 | (TC)12 | F: GATGGTAATGTGACGTGAGCAAAA R: CCTATTCTCATCGTTTAAAGCCCC | 247–287 | 58 | 20 | 6.084 | 0.987 | 0.859 | 0.870 | 0.014 | 17.607 | Ag01 [19] |
| Acg22 | (TC)11 | F: AACTTGTCTTATTGTGCACTTGCG R: ACATTTACGGCTAAACAGCATTCC | 184–210 | 58 | 8 | 3.092 | 0.956 | 0.701 | 0.713 | 0.017 | 14.456 | Ag05 [19] |
| Acg23 | (TG)12 | F: CAAGCGAAATAGATTCGTGGTCTT R: CTTCCATTTGGAGCCTTAAAACAC | 248–280 | 58 | 14 | 3.312 | 0.478 | 0.737 | 0.754 | 0.016 | 15.375 | Ag09 [19] |
| Ace1 | (TC)11n(TC)8ta (TC)6 | F: GCGCGCTCTCACTTTCTCTA R: CCACTTCTCCTCCTCGTCAC | 220–248 | 60 | 13 | 1.689 | 0.555 | 0.430 | 0.436 | 0.008 | 31.00 | FQ338662 [20] |
| Ace3 | (CT)12(CA)6 | F: TTCCTCAACAAAAACCCACC R: CCACGTCTCCCGAACACTAT | 210–240 | 60 | 11 | 3.786 | 0.935 | 0.760 | 0.792 | 0.039 | 6.16 | FQ335170 [20] |
| Ace27 | (CT)13(CA)7 | F: CCTTCTCACTTCACCCTCCA R: GAGGATGTGTGGCATTGTTG | 155–185 | 58 | 15 | 6.536 | 0.990 | 0.872 | 0.883 | 0.013 | 18.981 | FQ351578 [20] |
| Ace29 | (TC)26 | F: TGAATTTCTCCGGGACTTTG R: TCTCGGGAAATATCGACTTCA | 219–285 | 58 | 27 | 7.318 | 0.951 | 0.918 | 0.934 | 0.018 | 13.639 | FQ344263 [20] |
| Ace35 | (TC)20 | F: CGTGTGTGCCTTTGACCTTA R: ACTATTCAAGGAGGACGCGA | 194–200 | 58 | 4 | 1.682 | 0.681 | 0.431 | 0.459 | 0.060 | 3.917 | FQ334282 [20] |
| Ace37 | (GA)19 | F: CGGCAAGAACAACGAAGAAT R: AGCTGGAAGCTGATGACGAT | 119–137 | 58 | 10 | 1.402 | 0.376 | 0.307 | 0.315 | 0.005 | 49.75 | FQ351410 [20] |
| Act1 | (AG)5 | F: GGCCTGGATGTTGAGATAGC R: ATCAACTGACAACAGGCAACC | 122–134 | 60 | 4 | 1.846 | 0.654 | 0.483 | 0.541 | 0.106 | 2.108 | [20] |
| Act7 | (GT)5 | F: CAGCGACTAACACTGCATGA R: TATTGCTCAAAGGGAGCAGC | 239–245 | 59 | 4 | 1.640 | 0.476 | 0.414 | 0.406 | −0.020 | 1000 | [20] |
| Act9 | (TA)5 | F: TTTGTGGTTGGTTGAAGGTC R: AAAGGGTGAAGACGTGGATG | 204–212 | 59 | 5 | 1.529 | 0.605 | 0.366 | 0.362 | −0.011 | 1000 | [20] |
| Act12 | (AT)7 | F: CACGTACGCACGCATGTA R: TGAGTTCATTATCTTCGTGATTGA | 124–150 | 59 | 14 | 3.846 | 0.479 | 0.822 | 0.871 | 0.055 | 4.295 | [20] |
| Act15 | (TC)5 | F: TCGAAGTACACAATTTGCCA R: TTCTTGTCCTCTTTGGATTCG | 245–251 | 59 | 4 | 1.980 | 0.812 | 0.521 | 0.504 | −0.035 | 1000 | [20] |

**Table 2.** *Cont.*

| Locus | Repeat Motif | Primer Sequence (5′-3′) | Size Range | Tm (°C) | *Na* | *Ne* | *Ho* | *Hs* | *H′t* | *G′st*(Nei) | *Nm* | Primer Source |
|---|---|---|---|---|---|---|---|---|---|---|---|---|
| Act23 | (TA)5 | F: GGAGCTAGTCGAGGCATTTC<br>R: TTTGGGTTCAGAGCTTCACTC | 219–227 | 59 | 5 | 2.045 | 0.862 | 0.536 | 0.536 | −0.002 | 1000 | [20] |
| Act27 | (CT)7 | F: GCTTCTCTGCTGTTTGGTCC<br>R: TGTGCGAAACTCGCAAATTA | 217–227 | 60 | 6 | 1.649 | 0.288 | 0.423 | 0.452 | 0.064 | 3.656 | [20] |
| Act29 | (AT)6 | F: ATTCCCGACTGTCTCAAAGC<br>R: CTCTTTGATCCTTTGTTCTTTGG | 147–153 | 59 | 4 | 2.608 | 0.857 | 0.641 | 0.660 | 0.029 | 8.371 | [20] |
| Act30 | (TA)5 | F: GGTGGTTCGAACACTCCTCT<br>R: CATTCACTTACCAATTGCCC | 103–131 | 59 | 14 | 3.162 | 0.535 | 0.730 | 0.745 | 0.016 | 15.375 | [20] |
| Mean | | | | | 10.84 | 3.202 | 0.739 | 0.640 | 0.655 | 0.021 | 11.114 | |

*Na*: Number of different alleles; *Ne*: Number of effective alleles; *Ho*: Observed heterozygosity; *Hs*: Expected heterozygosity within populations; *H′t*: Corrected total expected heterozygosity; *G′st*(Nei): Corrected fixation index; *Nm*: Gene flow.

*2.3. Data Analysis*

For the polyploid characteristics, SSR genotype data were initially transformed into the GenoDive and STRUCTURE files using the Polysat package [21]. Genetic diversity per locus and population was evaluated through the following descriptive statistics; number of different alleles ($Na$), number of effective alleles ($Ne$), expected heterozygosity within populations ($Hs$), corrected total expected heterozygosity ($H't$), corrected fixation index ($G'st(Nei)$), total expected heterozygosity ($H_t$), and inbreeding coefficient ($Gis$) using GenoDive 2.0b27 [22]. Gene flow ($Nm$) was calculated using the formula: $Nm = (1 - G'st(Nei))/4\,G'st(Nei)$ [23]. GenoDive was also used in analysis of molecular variance (AMOVA) to investigate genetic variation among and within populations using 10,000 permutations.

Nei's genetic distance between populations was calculated by GenoDive, which was then used to construct the unweighted pair-group method with arithmetic means (UPGMA) tree using the NTSYS-pc 2.10s software [24,25]. Principal coordinate analysis (PCoA) was performed using Polysat based on Lynch's genetic distance among individuals and GenAlEx6.502 based on Nei's genetic distance between populations [21,26], respectively.

Population genetic structure was analyzed based on Bayesian clustering using STRUCTURE 2.3.4 [27,28]. For cluster values from K = 1 to K = 10, an admixture ancestry model and correlated allele frequency model were used to perform a Markov chain Monte Carlo simulation algorithm (MCMC). The length of the burn-in period was set to 100,000; MCMC after the burn-in period was set to 200,000, and for each K value, the calculation was repeated 10 times. The method from Evanno [29] was used to determine the optimal K value. The program STRUCTURE HARVESTER was used to calculate the optimal value of K using the deltaK criterion [30], then repeated sampling analysis was performed with CLUMPP 1.1.2 [31], and the inferred clusters were drawn as colored box plots with DISTRUCT1.1 [32].

To analyze the correlation between genetic distance and geographical distance or elevation difference, the Mantel test was performed by NTSYS-pc 2.10s using 1000 random permutations [33].

## 3. Results

*3.1. Population Genetic Diversity*

Amplification of 175 *A. cremastogyne* individuals representing 14 populations using 25 SSR primer pairs generated a total of 271 alleles (Table 2), for a mean of 10.84 alleles at each locus. Corrected total expected heterozygosity ($H't$) and observed heterozygosity ($Ho$) ranged from 0.315 to 0.934 and from 0.288 to 0.990, with means of 0.655 and 0.739, respectively. The locus that exhibited the highest $H't$ was Ace29 ($H't = 0.934$), which was also the locus that contained the largest number of alleles, 27; the number of effective alleles ($Ne$) was 7.318. Among all 25 SSR loci, Acg8, Acg23, Act12, Act27, and Act30 had lower $Ho$ than $H't$, whereas the rest had higher $Ho$ compared to $H't$. Overall, the mean of $Ho$ was higher than that of $H't$, indicating a heterozygote excess in the *A. cremastogyne* populations.

The genetic diversity of 14 *A. cremastogyne* populations is shown in Table 3. In the 14 populations, $Na$ and $Ne$ ranged from 4.16 to 7.80 and 2.842 to 3.962, with means of 5.83 and 3.37, respectively. Population F (Mianning) had the highest $Na$ ($Na = 7.80$), whereas population R (Yanting) had the lowest $Na$ ($Na = 4.16$). Population F (Mianning) also had the highest $Ne$ ($Ne = 3.962$), and population R (Yanting) had the lowest $Ne$ ($Ne = 2.842$). $Ho$ ranged from 0.706 to 0.800, and $Hs$ ranged from 0.593 to 0.682, with means of 0.739 and 0.630, respectively. Values for $Ht$ were identical to those for $Hs$. These 14 populations all had higher $Ho$ compared to $Hs$ and $Ht$. $Ho$ was lowest in population U (Muchuan) and highest in population T (Qionglai), whereas $Hs$ and $Ht$ were the lowest in population R (Yanting) but highest in population F (Mianning).

Thus, the overall genetic diversity in *A. cremastogyne* populations was relatively high and populations F (Mianning) and AA (Meigu) showed the highest diversity, followed by population B (Luding). By comparison, populations R (Yanting), I (Pingwu), and K (Jiange) showed a slightly lower level of genetic diversity.

**Table 3.** Genetic diversity of 175 *A. cremastogyne* trees representing 14 populations in Sichuan Province as detected by allele sizes at 25 simple sequence repeat loci.

| Population | *Na* | *Ne* | *Ho* | *Hs* | *Ht* | *Gis* |
|:---:|:---:|:---:|:---:|:---:|:---:|:---:|
| AA | 6.92 | 3.947 | 0.762 | 0.681 | 0.681 | 0.005 |
| AC | 5.68 | 3.301 | 0.724 | 0.623 | 0.623 | −0.036 |
| B | 6.28 | 3.615 | 0.711 | 0.643 | 0.643 | 0.020 |
| F | 7.80 | 3.962 | 0.794 | 0.682 | 0.682 | −0.038 |
| I | 5.36 | 2.905 | 0.735 | 0.608 | 0.608 | −0.055 |
| J | 5.04 | 3.139 | 0.758 | 0.615 | 0.615 | −0.083 |
| K | 5.68 | 3.266 | 0.709 | 0.605 | 0.605 | −0.029 |
| N | 6.72 | 3.450 | 0.712 | 0.606 | 0.606 | −0.037 |
| Q | 5.64 | 3.307 | 0.741 | 0.621 | 0.621 | −0.050 |
| R | 4.16 | 2.842 | 0.712 | 0.593 | 0.593 | −0.046 |
| S | 5.48 | 3.277 | 0.725 | 0.617 | 0.617 | −0.043 |
| T | 5.08 | 3.297 | 0.800 | 0.656 | 0.656 | −0.116 |
| U | 6.68 | 3.611 | 0.706 | 0.629 | 0.629 | −0.007 |
| V | 5.04 | 3.261 | 0.761 | 0.637 | 0.637 | −0.071 |
| **Mean** | 5.83 | 3.37 | 0.739 | 0.630 | 0.630 | −0.042 |

*N*a: Number of different alleles; *N*e: Number of effective alleles; *H*o: Observed heterozygosity; *H*s: Expected heterozygosity within populations; $H_t$: Total expected heterozygosity; *G*is: Inbreeding coefficient.

### 3.2. Population Genetic Differentiation

$G'_{ST}$, which is used to measure genetic differentiation among populations, ranged from 0.106 (Act1) and −0.035 (Act15) across the 25 loci, with a mean of 0.021 (Table 2). This indicates that the average genetic differentiation among 14 *A. cremastogyne* populations was 2.1%; i.e., 2.1% of the genetic variation existed among populations, whereas 97.9% of genetic variation existed within populations. Therefore, genetic variation within populations was the main source of variation. The average genetic flow *N*m in 25 SSR loci among 14 populations was 11.114 (Table 2), which was far greater than 1, consistent with the low level of genetic differentiation among populations, indicating that the genetic flow among 14 *A. cremastogyne* populations was relatively frequent. This is supported by the observation that *H*o was higher than *H*s or *H*t in all 14 populations. In other words, there was an excess of heterozygotes. We estimated the genetic differentiation coefficient among pairwise populations of *A. cremastogyne* (Table S1). The results showed that *F*st ranged from 0.012 to 0.037, and the average *F*st was also 0.021. Among pairwise combinations, the value of *F*st between populations R (Yanting) and V (Emeishan) was the largest (*F*st = 0.036), whereas that between populations AA (Meigu) and F (Mianning) was the smallest (*F*st = 0.021). Accordingly, *N*m ranged from 6.507 to 20.583, all greater than 1, suggesting a relatively high level of gene flow in pairwise populations, and it was difficult for populations to differentiate [34].

We performed AMOVA among and within *A. cremastogyne* populations using the 25 SSR loci (Table S2). The percentage of variation indicated that the genetic diversity was mostly within populations (97%), with only 3% occurring among populations. This result was consistent with *G*′st(*Nei*) = 0.021 calculated from F-statistics (Table 2). Both results indicated that the genetic variation in *A. cremastogyne* mainly occurs within populations.

### 3.3. Population Genetic Structure

The UPGMA tree constructed from the values of Nei's genetic distance showed that *A. cremastogyne* populations can be clearly divided into three clusters (AA and F, B, and others). Populations AA and F were from Liangshan Prefecture, population B was from Luding in Ganzi Prefecture, and all others were populations from the mountainous area around the Sichuan Basin and the central Sichuan hill area (Figure 2A). This indicates that the population genetic variation in *A. cremastogyne* exhibited a geographical distribution (Figure 1, Table 1). The third cluster can be further divided into two subgroups, one comprising AC, K, T, and N, and the other consisting of I, J,

Q, U, R, S, and V. However, these subgroups did not exhibit clear geographic differentiation (Figure 1, Table 1).

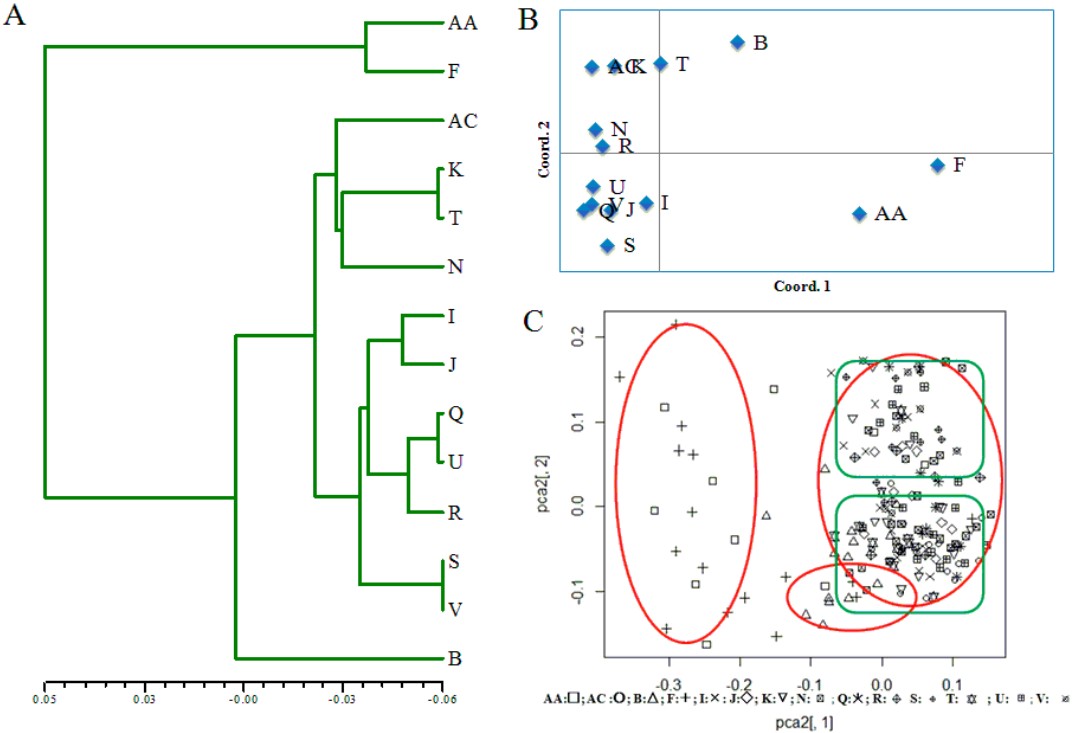

**Figure 2.** Population genetic structure. (**A**) Unweighted pair-group method with arithmetic means (UPGMA) cluster analysis of 14 *A. cremastogyne* populations based on Nei's genetic distance. (**B**) Principal coordinate analysis (PCoA) of 14 *A. cremastogyne* populations. (**C**) Principal coordinate analysis (PCoA) of 14 *A. cremastogyne* populations based on Lynch genetic distance between individuals.

The population structure analysis showed that at K = 3, ΔK was at the optimal value, indicating that the 14 *A. cremastogyne* populations could be divided into three clusters (Figure S1). The proportion of cluster membership of each individual from 14 populations is shown (Figure 3). At K = 3, based on the Q values, we graphed the proportion of cluster membership of each population (Figure 4). The graphs showed that the geographical distribution of original populations was relatively clear, where the Liangshan Prefecture populations (AA, F), Ganzi Prefecture population (B), and other populations exhibited clear differences. In the first cluster, original populations (red) had a larger proportion, especially in the Liangshan Prefecture populations. For populations from the mountain area around the Sichuan Basin and the central Sichuan hill area, the second (blue) and the third cluster (green) had a bigger proportion, with a very small proportion of the first cluster. This result is consistent with that of UPGMA clustering analysis. Similarly, the PCoA, which is based on either population or individual genetic distance, also showed that the 14 *A. cremastogyne* populations could be divided into three main groups (Figure 2B,C).

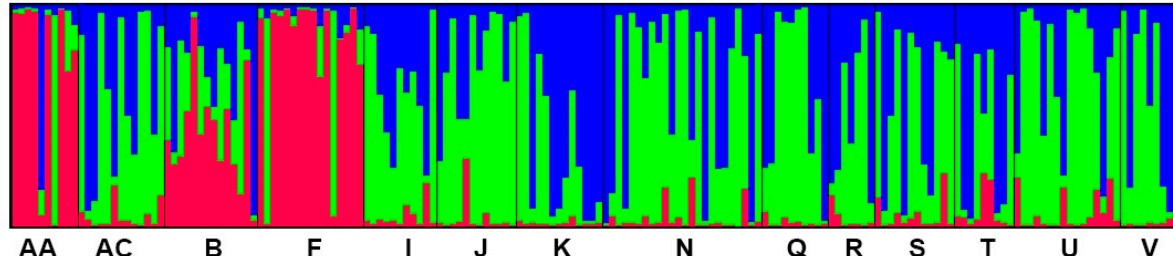

**Figure 3.** The proportions of cluster memberships at the individual level in the 14 *A. cremastogyne* populations based on STRUCTURE at K = 3. Each individual is represented by a single vertical bar, which is partitioned into three different colors. Each color represents a genetic cluster (see text for details), and the colored segments shows the individual's estimated ancestry proportion to each of the genetic clusters. The population codes are as in Table 1.

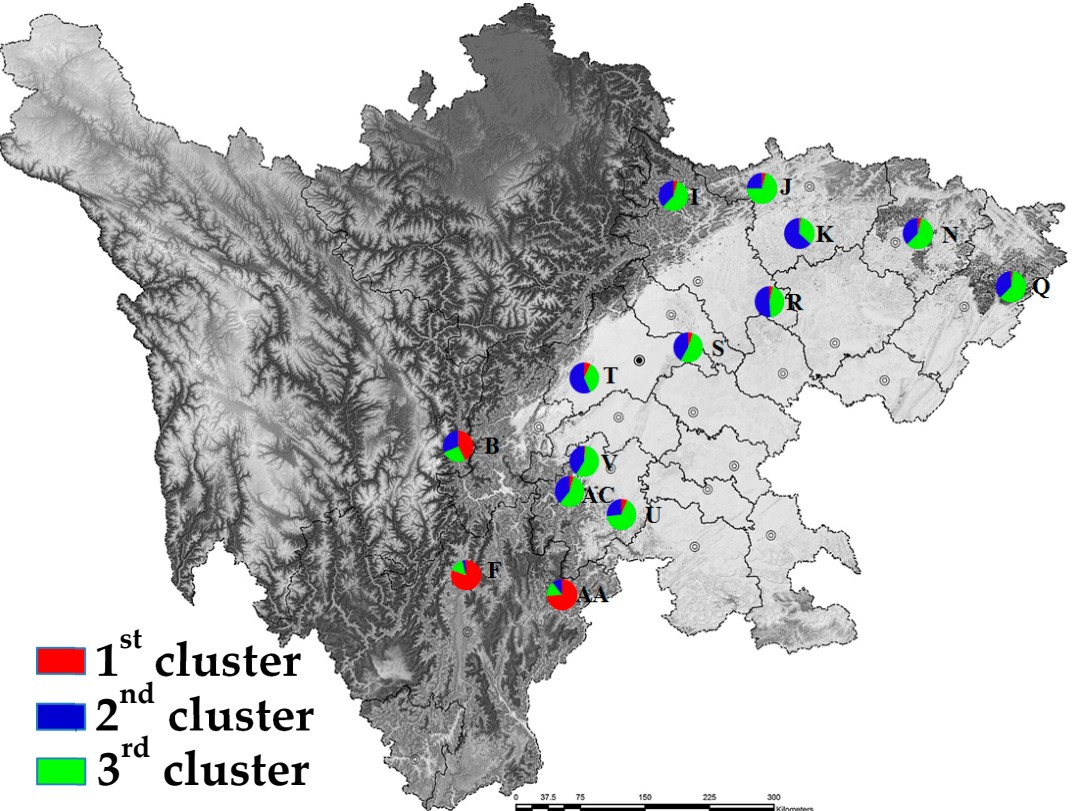

**Figure 4.** Mean proportions of cluster memberships of analyzed individuals in each of the 14 *A. cremastogyne* populations based on STRUCTURE at K = 3.

We further analyzed the correlation between Nei's genetic distance among populations and geographical distance as well as elevation difference using the Mantel test. The correlation coefficient between genetic distance and geographical distance ($r = 0.187$ ($p = 0.0759$)) indicated that the correlation between genetic and geographical distance was not significant, and therefore did not exhibit a clear geographic distribution (Figure 5A). However, the Mantel test between genetic distance and elevation showed that the population genetic variation in *A. cremastogyne* exhibited an significant elevation distribution [$r = 0.721$ ($p = 0.0040$)] (Figure 5B).

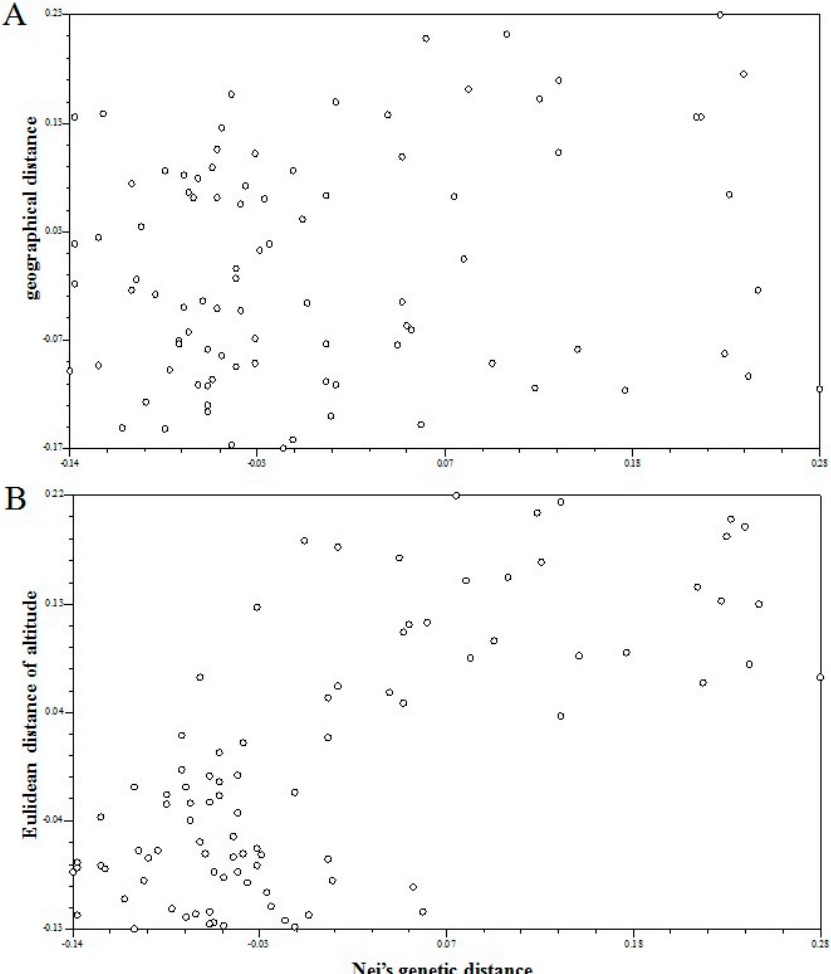

**Figure 5.** Mantel test. (**A**) Mantel between Nei's genetic distance and geographical distance of 14 *A. cremastogyne* populations. (**B**) Mantel between Nei's genetic distance and Euclidean distance of elevation for 14 *A. cremastogyne* populations.

## 4. Discussion

### 4.1. Polyploidy and Data Analysis in A. cremastogyne

In early SSR-related research, *A. glutinosa* was studied as diploid. Later, Lepais et al. [13] and Mandák et al. [35] found tetraploid populations (2n = 4× = 56) in Africa and Europe, respectively. The two tetraploid populations found by Mandák et al. [35] were located in the Iberian Peninsula and Dinaric Alps, respectively. Further analysis by the same laboratory proposed that these two tetraploid populations were new tetraploid species that were closely related to *A. glutinosa* and thus were subsequently named as *A. lusitanica* and *A. rohlenae* [36].

Ren et al. [37] and Yang et al. [38] conducted karyotype studies and found that *A. cremastogyne* had 56 chromosomes, and the basic number of the alder is x = 14 according to Hong et al. [39] or x = 7 based on Murai et al. [40]. Therefore, *A. cremastogyne* is possibly tetraploid or octoploid. Karyograms showed that chromosomes with the same structures were mostly in pairs [40], indicating the possibility that *A. cremastogyne* was undergoing or approaching the completion of diploidization. However, the obtained SSR genotype data revealed the characteristics of a tetraploid, so this study performed SSR data analysis in *A. cremastogyne* utilizing the polyploid analysis software Polysat and Genodive [21,22], combined with STRUCTURE, GenAlEx, and NTSYS software. Of course, if 2–3 alleles were detected in a tetraploid, it would be difficult to precisely identify the genotype of the individuals. For example, if

the genotype was AB, then the individual genotype could not be distinguished from ABBB, AABB, and AAAB. Thus, the population genetic diversity could be presumably underestimated [21].

*4.2. Population Genetic Structure and Geographical Variation in* A. cremastogyne

Overall, the 14 *A. cremastogyne* populations all have relatively high levels of genetic diversity ($N$a = 5.83, $H$t = 0.630), with the Liangshan Prefecture populations (Meigu, Mianning) and Ganzi population (Luding) having the highest level of genetic diversity. Based on the SSR data, the genetic diversity of *A. cremastogyne* is higher than that of *A. maritima* ($N$a = 4.70, $H$e = 0.45) [11] and *A. incana* ($N$a = 3.31, $H$e = 0.386) [41]. Meanwhile, the genetic diversity of *A. cremastogyne* is similar to that of the whole European population ($N$a = 6.70, $H$e = 0.643) [35], the Irish, Scottish, and French population ($N$a = 6.61, $H$e = 0.64) [17], as well as that of *A. glutinosa* population located at the border of Belgium, Luxembourg, and France ($N$a = 7.34, $H$e = 0.64) [41]. In addition, the genetic diversity of *A. cremastogyne* in this study was also consistent with that of species that are perennial ($H$e = 0.68), outcrossed ($H$e = 0.65), regionally distributed ($H$e = 0.65), and wind- or water-dispersed ($H$e = 0.61), from a study by Nybom et al. [42] based on SSR data. Also, the genetic diversity of *A. cremastogyne* was higher than that of species with a narrow distribution ($H$e = 0.56), early succession ($H$e = 0.42), and endemism ($H$e = 0.46) [42].

The genetic differentiation among *A. cremastogyne* populations was very low ($F$st = 0.021), lower than that of *A. maritima* ($F$st = 0.107) [11] and *A. incana* ($F$st = 0.0836) [41] populations, but similar to that of *A. glutinosa* populations at the border of Belgium, Luxembourg, and France ($F$st = 0.015) [35]. Wright [43] considered values of $F$st in the ranges of 0–0.05, 0.05–0.15, 0.15–0.25, or > 0.25, which represented low, moderate, high, and very high genetic variation among populations, respectively. Thus, the genetic differentiation among *A. cremastogyne* populations was low, and variation mostly existed within populations. This indicates that gene flow among *A. cremastogyne* populations was relatively frequent, preventing genetic differentiation among populations [34], consistent with the gene flow value ($N$m = 11.114).

For population classification, the *A. cremastogyne* populations can be divided into three clusters: Liangshan Prefecture populations, Ganzi Prefecture populations, and those located in the mountainous area around Sichuan Basin and the central Sichuan hill area. This suggests that the population genetic variation in *A. cremastogyne* has certain characteristics of geographical distribution. However, further classification within the populations located in the mountain area around Sichuan Basin and the central Sichuan hill area did not correlate with the geographical locations. This is possibly due to the large residential population in this area and extensive human disturbance of *A. cremastogyne* populations such as the continuous reforestation practices. It is also possible that *A. cremastogyne* prefers river bank habitats, and its nutlets (seeds) have wings and are very lightweight; therefore, these are easily affected by regional water systems such as the Minjiang River, Fujiang River, and Jialingjiang River (Figure 1). The population genetic structure of *A. cremastogyne* exhibits significant isolation by distance (IBD) caused by elevation gradient, represented by the fact that as the elevation difference increased among populations, the genetic distance became greater.

The pattern of population genetic variation and differentiation in *A. cremastogyne* mentioned above may be tightly correlated with the rise of the Qinghai–Tibetan Plateau (The Liangshan and Ganzi areas in this study belong to the southeastern border of the Qinghai–Tibetan Plateau). A study by Chen et al. [1] considered that the Betulaceae plants possibly originated from areas near the tropics that are subjected to slight seasonal drought. Chen et al. [1] believed that the Central China area, which centers on Sichuan Province, was the center of origin and early differentiation of the Betulaceae; their ancestors lived as early as the Santonian age of the Cretaceous period or even earlier. The earliest fossil record revealed that alder had diverged from their ancestors by the middle Eocene [1]. In the stratum with a geological age of 34.6 ± 0.8 Ma (paleoelevation 2910 ± 910 m) in Mangkang, Tibet, alder fossils were discovered, dating back to the Eocene–Oligocene Transition [44]. An analysis combining growth traits and ecological gradients showed that areas of Qionglai, Yaan, Emeishan, and Dujiangyan

were considered to be the alder origin center. [45]. Thus, the origin and dispersal of *A. cremastogyne* could be accompanied by the uplift of the Qinghai–Tibetan Plateau. Wang et al. [45] believed that the provenance growth characteristics of *A. cremastogyne* exhibited longitude and latitude variation. Within the natural distribution areas of *A. cremastogyne*, as the longitude decreases, elevation tends to increase; therefore, elevation is possibly a positive selection factor for genetic diversity in *A. cremastogyne*. Meanwhile, elevation could also be a main impact factor for the population genetic structure and growth traits of *A. cremastogyne*. Moreover, this will necessarily require further investigation.

In addition, among *A. ferdinandi-coburgii*, *A. nopalensis*, and *A. lanata*, some have different ecological niches from *A. cremastogyne*, whereas some show niche overlap with *A. cremastogyne*. The differentiation of these alder species in this area also requires further investigation.

*4.3. Genetic Improvement of* A. cremastogyne

The natural environmental conditions of *A. cremastogyne* habitats are complicated as these involve a vast range of geological and climate types. Also, because of expanding human activities, the germplasm resources of *A. cremastogyne* are facing continuous destruction. Therefore, selecting a number of core populations for in situ conservation is a suitable approach for the protection of *A. cremastogyne* and its population genetic diversity [46]. *A. cremastogyne* populations F, AA, and B exhibited the greatest genetic diversity. Thus, these populations should be considered priorities for in situ conservation. In addition, establishing ex situ conservation such as a breeding population, which shows the capacity to have large diversity in Cubry et al [17], should also be considered to meet plant breeding needs.

To date, the importance of *A. cremastogyne* as a pioneer tree species has received little attention. In studies of *A. maritima*, it was considered that in earlier stages, *A. maritima* was widely distributed across the North American continent; as an early succession tree species, its ability to fix nitrogen created conditions for the establishment of its successors. Together with its preference for light, high outcrossing rate, and low seed germination under natural conditions, it led to a fragmented distribution [11,12,47,48]. Further studies on the ecological function, population succession, mating system, and the mechanism of seed dispersal, as well as revealing their ecological benefits and more detailed distribution pattern are warranted for the genetic improvement of *A. cremastogyne*.

## 5. Conclusions

This is first study to utilize SSR markers to analyze the genetic diversity and genetic structure of populations. Our study showed that the 14 *A. cremastogyne* populations have a relatively high level of genetic diversity. Gene flow among *A. cremastogyne* populations was relatively frequent, which weakened the variation among populations. There was little genetic differentiation among the *A. cremastogyne* populations; the majority of variation was within populations. These 14 *A. cremastogyne* populations could be divided into three clusters and exhibited characteristics of geological distribution. Furthermore, the present study showed that besides SSR primers from other *Alnus* species, the GenBank EST database of *A. glutinosa* and the unpublished transcriptome data of *A. cremastogyne* can also be applied for the screening of SSR primers. This will provide sufficient microsatellite markers for studying genetic variation in alder, including *A. cremastogyne*. Taken together, the findings of our study may potentially assist in the conservation management and genetic improvement of *A. cremastogyne*.

**Supplementary Materials:** The following are available online at http://www.mdpi.com/1999-4907/10/3/278/s1, Figure S1. Relationship between the rational cluster K and estimated value ΔK, Table S1: Values of $F$st (below the diagonal) and $N$m (above the diagonal) for pairwise comparison of *A. cremastogyne* populations estimated estimated using 25 simple sequence repeat loci, Table S2: Molecular variance analysis within and among *A. cremastogyne* populations estimated using 25 SSR loci.

**Author Contributions:** X.-Y.K. and H.-Y.G. conceived and designed the experiments; Z.C. and H.-B.Y. collected the plant samples; Z.-L.W. and Z.H. performed the experiments and data analysis; and H.-Y.G. and Z.-L.W. wrote the paper. All authors read and approved the final manuscript.

**Funding:** This research was supported by the Sichuan Science and Technology Program, grant nos. 2016NYZ0035-02 and 2017JY0278.

**Conflicts of Interest:** The authors declare no conflict of interest.

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
