# Peer review of "Genetic Diversity and Population Structure of Alnus cremastogyne as Revealed by Microsatellite Markers"

_forests, doi:10.3390/f10030278_

Round 1

Reviewer 1 Report

Genetic Diversity and Population Structure of Alnus cremastogyne as Revealed by Microsatellites Markers

This paper investigates the genetic diversity and population structure of Alnus cremastogyne in China. It is not clear if the samples were collected across the entire range, but the impression is that they have been. The populations were analysed using 25 SSRs. Relatively high levels of diversity were uncovered with very low levels of population structure or differentiation. The only significant differentiation was with altitude. This may be to do with the lack of independence of the altitude data – i.e. similar altitudes will be subsets of the geographical data. Overall the data corresponded well with other published studies and is a good start on pre-breeding for this species.

Overall the paper reports on a good pre-breeding study that has relevance for forestry and also a wider community of ecologists and evolutionary biologists.

Points to address:

P2, l 69. An important reference is missing here (Cubry et al 2015). This is a study on Alnus glutinosa in Ireland and it should be include. This is relevant for comparisons of heterozygosity but also as it compares wild populations with an ex situ breeding population.

Cubry P, Gallagher E, O’Connor E, Kelleher C (2015) Phylogeography and population genetics of black alder (Alnus glutinosa (L.) Gaertn.) in Ireland: putting it in a European context. Tree Gene Genomes 11:99

P2, l85, - was only FAM used. This limits potential multiplexing?

P3, Figure 1. It would be useful to give the reader an indication of the distribution range of the species so that the populations sampled can be put into that context. It is unclear if the authors sampled across the range or a subset of the range and this will influence the interpretation of the results.

P6, 3. Results. Did the authors assess the PIC differences between the different SSR sources, e.g. Ace29 has the greatest number of alleles and comes from A. cremastogyne? On casually looking I cannot see any patterns, but it would be good to check this. Cross-species SSRs are very useful and it would be of interest to note this.

P6, l148-151. These differences are not significant. The range from lowest (0.593) to highest (0.682) is only (0.089)? All values are relatively high.

P9, l218. Is the mantel test with altitude giving a bias as it is giving a subset of the geographical data? It looks like from Figure 5, that the altitude effect is a combination of altitude and geography.

P11, l236-249. Another option to polyploidy is to treat the data as dominant data and analyse it similar to an AFLP matrix.

P11, l 256. The Irish study (Cubry et al 2015) shows similar levels He of 0.64 and Na of 6.61. This should be included and referenced here.

P13, l 312. The Cubry et al (2015) study shows very high levels of variation in the breeding population, which is a subset of the natural population. It shows the capacity to have large diversity in an ex situ population.

Some specific comments are listed below:

P1, l 44, replace “of” with “in” to read …”…species in China …”

P1, l 39, add “e” to read “…. Betulaceae family; ….”

P2, l 50 and l 52, delete “s” to read “…genetic variation.”

P2, l 59, Edit to read “….. markers occur widely, are highly polymorphic ….”

P2, l 71, insert “s” to read “…molecular markers.”

P2, l 75-77, check wording of this sentence.

P2, l 78. I assume leaf samples were taken. State if it was leaf samples.

P2, l79, insert “gel” to read “…silica gel, transported …… ”

P2, l 85, replace “former” with “forward” to read “..forward primer ..”

P6, l 132, replace “on” with “at” to read “..alleles at each locus.”

P6, l 136, add “s” to read “..alleles (Ne) …”

P6, l 138, add “a” to read “…indicating a heterozygous …”

P11, l 238, insert space to read “..genus was   …”

P11, l241-245. Check the wording of this sentence.

P12, l 260, add “s” to read “..distributions ..”

P12, l 262, reword to read “… populations was very low (Fst = 0.021), ….”

P12, l 303 and 306. There is no need to repeat the authority here as it has already been used at the beginning of the paper.

P13, l327, insert “s” to read “…SSR markers …..”

Author Response

Dear Mr./Ms., 

We are truly grateful for your comments, constructive criticism, and thoughtful suggestions. Based on these comments and suggestions, we have made careful modifications to the original manuscript. All changes are marked with the Track Changes function in Microsoft Word. We have also consulted a professional English editing service to polish our paper before resubmission. We hope the new manuscript will meet the magazine’s standards. Below you will find our point-by-point responses to the comments and suggestions. 

Point 1: 

P1, l 44, replace “of” with “in” to read …”…species in China …” 

P1, l 39, add “e” to read “…. Betulaceae family; ….” 

P2, l 50 and l 52, delete “s” to read “…genetic variation.” 

P2, l 59, Edit to read “….. markers occur widely, are highly polymorphic ….” 

P2, l 71, insert “s” to read “…molecular markers.” 

P2, l 75-77, check wording of this sentence. 

P2, l 78. I assume leaf samples were taken. State if it was leaf samples. 

P2, l79, insert “gel” to read “…silica gel, transported …… ” 

P2, l 85, replace “former” with “forward” to read “..forward primer ..” 

P6, l 132, replace “on” with “at” to read “..alleles at each locus.” 

P6, l 136, add “s” to read “..alleles (Ne) …” 

P6, l 138, add “a” to read “…indicating a heterozygous …” 

P11, l 238, insert space to read “..genus was …” 

P11, l241-245. Check the wording of this sentence. 

P12, l 260, add “s” to read “..distributions ..” 

P12, l 262, reword to read “… populations was very low (Fst = 0.021), ….” 

P12, l 303 and 306. There is no need to repeat the authority here as it has already been used at the beginning of the paper. 

P13, l327, insert “s” to read “…SSR markers …..” 

Response 1: We have made careful revisions in response to all your suggestions. 

Point 2: P2, l 69. An important reference is missing here (Cubry et al 2015). This is a study on Alnus glutinosa in Ireland and it should be include. This is relevant for comparisons of heterozygosity but also as it compares wild populations with an ex situ breeding population

 P11, l 256. The Irish study (Cubry et al 2015) shows similar levels He of 0.64 and Na of 6.61. This should be included and referenced here. 

P13, l 312. The Cubry et al (2015) study shows very high levels of variation in the breeding population, which is a subset of the natural population. It shows the capacity to have large diversity in an ex situ population. 

Response 2: We now cite Cubry’s study in three places (Line 69, Line 260, and Line 320) in our modified manuscript. 

Point 3: P2, l85, - was only FAM used. This limits potential multiplexing? 

Response 3: Yes, only FAM was used in our study. We regret that the convenience and low price of multiple fluorescents and subsequent multiplexing were not considered when performing PCR amplification. 

Point 4: P3, Figure 1. It would be useful to give the reader an indication of the distribution range of the species so that the populations sampled can be put into that context. It is unclear if the authors sampled across the range or a subset of the range and this will influence the interpretation of the results. 

Response 4: A. cremastogyn (also known as the Sichuan alder) is naturally distributed in Sichuan Province and adjacent areas. In 2011, we began to survey and collected natural alder resources throughout the area. As mentioned in this manuscript, they were also influenced by extensive human disturbance. So we sampled across the entire known range. 

Point 5: P6, 3. Results. Did the authors assess the PIC differences between the different SSR sources, e.g. Ace29 has the greatest number of alleles and comes from A. cremastogyne? On casually looking I cannot see any patterns, but it would be good to check this. Cross-species SSRs are very useful and it would be of interest to note this. 

Response 5: For the polyploid characteristics, SSR genotype data were initially analyzed using Polysat package and GenoDive for polyploid species, but we are sorry that neither of these have a function for calculating PIC. We referred to some papers that did list PIC, and we found that Cervus could be used to calculate PIC, but this software is only used for diploid species. 

However, we are grateful to have been reminded to check for the effect of three different sources of SSR on evaluation of genetic diversity in A. cremastogyne. We may address this in another short manuscript in the future. 

Point 6: P6, l148-151. These differences are not significant. The range from lowest (0.593) to highest (0.682) is only (0.089)? All values are relatively high. 

Response 6: About this point, we changed the wording of this sentence to “populations F (Mianning) and AA (Meigu) showed the highest diversity, followed by population B (Luding). By comparison, populations R (Yanting), I (Pingwu), and K (Jiange) showed a slightly lower level of genetic diversity”. 

Point 7: P9, l218. Is the mantel test with altitude giving a bias as it is giving a subset of the geographical data? It looks like from Figure 5, that the altitude effect is a combination of altitude and geography. 

Response 7: As mentioned in Response 4, we collected samples across the entire natural range in A. cremastogyne, so the geographical data could not possibly result in a bias. But in the areas investigated, as the longitude decreased, elevation tended to increase and geography also tended to change, so we here concluded that elevation may be a main factor affecting the genetic differentiation of A. cremastogyne. 

Point 8: P11, l236-249. Another option to polyploidy is to treat the data as dominant data and analyse it similar to an AFLP matrix. 

Response 8: We also treated the data in this way. Our SSR genotype data were transformed into dominant data using the Polysat package, and then genetic diversity was evaluated using GenAlEx6.502, but showing very low genetic diversity. We think this may come from the limited quantity of dominant data used here. 

Once again, thank you very much for your comments and suggestions. 

Sincerely yours, 

Hong-ying Guo

Reviewer 2 Report

Review of Alnus cremastogyne genetic diversity manuscript for Forests. 

General comments on the manuscript: 

The analysis was based on the assumption that Alnus cremastogyne is a tetraploid species, yet this is not mentioned until very late in the text (line 237).  It should be mentioned at the beginning so that readers understand the choice of software and the methods of analysis. 

Reference numbers should be given immediately after the authors [1] rather than at the end of a sentence. 

The taxonomic authority should be given only once (Alnus cremastogyne Burk. and Alnus Mill.).  After that, only the scientific name is needed. 

The tables and figures should be self-explanatory.  The captions should include additional information, and terms should be defined in footnotes. 

The "number of effective alleles" is a term that may be unfamiliar to many readers.  For this reason, the term should be defined, how it is calculated should be stated, and how it is interpreted in the context of this study should be mentioned. 

From http://www.uwyo.edu/dbmcd/molmark/waap.html

AE is the effective number of alleles (at the level of the OTUs we are examining).  Verbally, this measure is the number of equally frequent alleles it would take to achieve a given level of gene diversity.  That is, it allows us to compare populations where the number and distributions of alleles differ drastically.  The formula is:

where Dj is the gene diversity of the jth of r loci.  Note that we calculate the OTU-level AE by averaging over the AE calculated locus-by-locus rather than by calculating a mean gene diversity and then calculating AE from that.  The graph below shows why: AE is a nonlinear function of the gene diversity (Hexp), which brings into play Jensen’s inequality [the expectation of a function ≠ the function of the expectations for nonlinear curves; see Ruel and Ayres (1999)].  Here, because the curve is concave up, the AE we compute will be greater than if we calculated it from the overall gene diversity.

Comments and corrections with line numbers shown. 

Line 18.  population genetic variation in A. cremastogyne using 175 trees sampled from 14 populations native to Sichuan province of China using 25 simple sequence repeat (SSR) markers. 

Line 24.  What is "mean genetic variation parameter"? 

Line 26.  delete "14" 

Line 28 population, and the other

Line 29.  some geographica differentiation. 

Line 31.  Replace "altitude" with "elevation" throughout the manuscript. 

Line 36.  contains non-leguminous

Line 39.  Betulaceae family  (note spelling; family name NOT in italics). 

Line 40.  Eurasia and North America, and is rarely seen in

Line 43.  Among these four species, A. cremastogyne is the most important endemic Chinese species and its most widely studied. 

Line 46.  high seed yield 

Line 48.  ecological reconstruction 

Line 50  genetic variation. 

Line 52.  genetic variation … for phenotyping [3-6]. 

Line 54.  Zhuo et al. [7] who used random amplified polymorphic DNA (RAPD) markers to study genetic variation 

Line 55.  Rao et al. [8] is the only study that developed simple sequence repeat (SSR) markers for Alnus.  They used transcriptome data from three species: 

Line 57.  Population genetic variation in A. cremastogyne lacks supporting data at the molecular level, which in turn limits the species' protection and utilization. 

Line 59.  SSR markers occur abundantly in plant genomes, and many are highly polymorphic 

Line 60.  population structure and evolutionary history.  The earliest report by Zhuk et al. (2008) found that 8 of 15 birch SSR primer pairs yielded amplification products in 

Line 63.  Lance et al. [10] designed 19 primer pairs from an enriched library of A. maritima and used them to amplify A. maritima and A. serrulata. 

Line 67.  SSR technologies were applied … and A. incana and the population genetic diversity, genetic structure, evolution and phylogeny were gradually elucidated 

Line 72.  This study aims to facilitate the genetic improvement and utilization of A. cremastogyne

Line 74.  Population sampling 

Line 75.  a mountainous area around the Sichuan basin, the central Sichuan hill area, an alpine valley region of western Sichuan, and the mountainous area of southwestern Sichuan.  Trees in the study areas were typical natural secondary forests (Fig. 1, Table 1), from which we sampled 175 individuals representing 14 populations.  The sampled trees …

Line 78.  Leaf samples were collected, dried with silica, transported …

Line 80.  2.2  DNA extraction and amplification 

Line 82.  A total of 25 primer pairs (Table 2) for highly polymorphic loci were selected from three sources:  published papers for Alnus species [  ], the GenBank EST database …

Line 85.  The forward primer was fluorescently labeled with 6-FAM. 

Line 87.  Figure 1.  Locations in Sichuan province of the 14 sampled populations of Alnus cremastogyne

Line 88.  Table 1.  Location and number of trees sampled in 14 populations of Alnus cremastogyne in Sichuan province.  [The column heading should show sample size or number of trees sampled.  "Size" is too vague]. 

Line 95.  Then, the PCR products were submitted for capillary electrophoresis on an ABI … and the fragment lengths were determined … State College, PA, USA.  [If post-PCR multiplexing was used, it should be mentioned here]. 

Line 100.  Table 2.  Characterization of 25 simple sequence repeat loci in Alnus cremastogyne based on 175 trees representing 14 populations in Sichuan province. 

Line 101.  Many words in this line should start with lower case letters. 

Line 104.  delete "the" 

Line 110.  If you calculate G'ST, the formula should include it rather than FST 

Line 113.  Nei's genetic distance between populations (is there a symbol?) 

Line 115.  was performed using Polysat based on  

Line 119.  For cluster values of K = 1 to K = 10,

Line 121.  burn-in period was set to 100,000; MCMC after the burn-in period was w35 5o 200,000, and for each K value the calculation was repeated 10 times.  The method of Evanno [28] was used …

Line 127.  elevation difference, the Mantel test was performed by NTSYS-pc2.10s 

Line 131.  Amplification of 175 A. cremastogyne individuals representing 14 populations using 25 SSR primer pairs generated a total of 271 alleles (Table 2), for a mean of 10.84 alleles at each locus.     

Line 133.  0.934 and from 0.288 to 0.990, with means of 0.665 and 0.739, respectively. 

Line 136.  delete "detected" 

Line 137.  deled "of the loci" 

Line 138.  excess in the A. cremastogyne populations, 

Line 143.  Ho ranged from 0.706 to 0.800, and Hs ranged from 0.593 to 0.682.  Values for Ht were identical to those for Hs. 

Line 148.  was relatively high and populations F (Mianning) and AA (Meigu) had the greatest diversity … In contrast, populations …

Line 154.  Table 3.  genetic diversity in 175 Alnus cremastogyne trees representing 14 populations in Sichuan province as detected by allele sizes at 25 simple sequence repeat loci. 

Line 155.  Readers should be referred to the map or Table 1 for the locations of the populations.  ALL terms should be defined in the footnote. 

Line 158.  ranged from 0.106 (Act1) to -0.035 (Act15) over the 25 loci, with a mean of 0.021. 

Line 161.  populations.  Therefore, genetic variation within populations was the main soruce of variation.  The average

Line 165.  This is supported by the observation that Ho was higher than Hs or Ht in all 14 populations.  In other words, there was an excess of heterozygotes. 

Line 168.  delete "with differences among pairwise populations observed," 

Line 170.  Among pairwise comparisons, the value

Line 174.  delete "and it was difficultu for populations to differentiate due to genetic drift".  I think the authors mean to say gene flow through pollen or seeds.  Genetic drift is random change in allele frequency due to small sample size, and is not the case in this study. 

Line 176.  Table S1.  Values of Fst (below the diagonal) and Nm (above the diagonal) for pairwise comparison of A. cremastogyne populations estimated using 25 simple sequence repeat loci. 

Line 177.  Delete.  It is not needed if the Table caption is expanded. 

Line 181.  calculated from F-statistics … genetic variation in A. cremastogyne mainly occurs within populations. 

Line 188.  Prefecture, and all others

Line 189.  genetic variation in

Line 192.  geographic differentiation 

Line 194.  that at K=3, … indicating that the 14 A. cremastogyne populations could be divided … 

Line 196.  is shown (Figure 4).  At K=3 

Line 200.  In the first cluster,

Line 204.  that of the UPGMA clustering analysis 

Line 205.  distance, also showed that the 14 A. cremastogyne populations could be divided into three main groups 

Line 211.  individuals. 

Line 212.  Figure 3 is not needed.  It could be deleted or moved to a supplementary figure. 

Line 214.  among populations and geographical distance as well as elevation difference using the Mantel test.  The correlation coefficient between genetic distance and geographical distance [r = 0.187 (p=0.0760] indicated that

Line 218.  replace "altitude" with "elevation"

Line 226.  In early SSR-related research 

Line 228.  Mandak et al [34] were located in the Iberian Peninsula 

Line 236.  delete "At the chromosome level" 

Line 237.  56 chromosomes and the basic number of the genus Alnus is x = 14 according to

Line 240.  delete "of two" 

Lines 239-248.  I assume that these are observations of chromosomes during meiosis? 

Line 249.  geographical variation in 

Lines 250-261.  Please note that comparisons are possible if the SAME SSR markers are used in the various studies.  Otherwise the comparisons become more general. 

Line 257.  in this study 

Line 258.  that are perennial.

Line 260  species with a narrow distribution 

Line 262.  was relatively low (FST

Line 264.  similar to        delete "located"     

Line 265.  Wright [42] considered values of FST in the ranges of 0 – 0.05, … or > 0.25 to represent low …

Line 268.  indicates that gene flow

Lines 265-270.  Also consider/discuss the origins of the populations and the reforestation practices used in the area.  

Line 272.  populations, and those located in the mountainous areas around the Sichuan basin …

Line 279.  seeds have small wings (?) 

Line 281.  change "altitude" to "elevation". 

Line 282.  change "altitude distance" to "elevation difference" 

Line 285.  The Liangshan and Ganzi areas in this study belong to the southeastern border of the Qinghai-Tibetan Plateau. 

Line 290.  the Cretaceous period (?)

Line 300.  delete "the distribution of" 

Line 307.  Birch (Betula) and hazelnut (Corylus) are also in the family Betulaceae, and some species are native to the areas investigated.  It would be appropriate to mention them. 

Line 313.  A. cremastogyne and its population genetic diversity. 

Line 314.  exhibited the greatest genetic diversity

Line 315.  considered priorities 

Line 316. considered to meet plant breeding needs. 

Line 318.  as a pioneer tree species has received little attention. 

Line 321.  establishment of its successors.  Together with its preference for light … it led to a fragmented distribution …

Line 327.  first study to utilize SSR markers to analyze

Line 328.  delete "A. cremastogyne

Line 330.  There was little genetic variation among the A. cremastogyne populations; the majority of the variation was within populations. 

Line 336.  genetic variation 

Lines 339, 341.  estimated using 25 SSR loci. 

References.  Most words in the article titles should start with lower case letters. 

Author Response

Dear Mr./Ms., 

We are truly grateful for your comments, constructive criticism, and thoughtful suggestions. Based on these comments and suggestions, we have made careful modifications to the original manuscript. All changes are marked with the Track Changes function in Microsoft Word. We have also consulted a professional English editing service to polish our paper before resubmission. We hope the new manuscript will meet the magazine’s standards. Below you will find our point-by-point responses to the comments and suggestions. 

Point 1: Reference numbers should be given immediately after the authors [1] rather than at the end of a sentence; 

The tables and figures should be self-explanatory. The captions should include additional information, and terms should be defined in footnotes. 

Line 18. population genetic variation in A. cremastogyne using 175 trees sampled from 14 populations native to Sichuan province in China with 25 simple sequence repeat (SSR) markers. 

Line 26. delete "14" 

Line 28 population, and the other 

Line 29. some geographica differentiation. 

Line 31. Replace "altitude" with "elevation" throughout the manuscript 

Line 39. Betulaceae family (note spelling; family name NOT in italics). 

Line 40. Eurasia and North America, and is rarely seen in 

Line 43. Among these four species, A. cremastogyne is the most important endemic Chinese species and its most widely studied. 

Line 46. high seed yield 

Line 48. ecological reconstruction 

Line 50 genetic variation. 

Line 52. genetic variation … for phenotyping [3-6]. 

Line 54. Zhuo et al. [7] who used random amplified polymorphic DNA (RAPD) markers to study genetic variation 

Line 55. Rao et al. [8] is the only study that developed simple sequence repeat (SSR) markers for Alnus using transcriptome data from three species: 

Line 57. Population genetic variation in A. cremastogyne lacks supporting data at the molecular level, which in turn limits the species' protection and utilization. 

Line 59. SSR markers occur abundantly in plant genomes, and many are highly polymorphic 

Line 60. population structure and evolutionary history. The earliest report by Zhuk et al. (2008) found that 8 of 15 birch SSR primer pairs yielded amplification products in 

Line 63. Lance et al. [10] designed 19 primer pairs from an enriched library of A. maritima and used them to amplify A. maritima and A. serrulata. phylogeny were gradually elucidated 

Line 72. This study aims to facilitate the genetic improvement and utilization of A. cremastogyne. 

Line 74. Population sampling 

Line 75. a mountainous area around the Sichuan basin, the central Sichuan hill area, an alpine valley region of western Sichuan, and the mountainous area of southwestern Sichuan. Trees in the study areas were typical natural secondary forests (Figure 1, Table 1), from which we sampled 175 individuals representing 14 populations. The sampled trees … 

Line 78. Leaf samples were collected, dried with silica, transported … 

Line 80. 2.2 DNA extraction and amplification 

Line 82. A total of 25 primer pairs (Table 2) for highly polymorphic loci were selected from three sources: published papers for Alnus species [ ], the GenBank EST database … 

Line 85. The forward primer was fluorescently labeled with 6-FAM. 

Line 87. Figure 1. Locations in Sichuan province of the 14 sampled populations of Alnus cremastogyne. 

Line 88. Table 1. Location and number of trees sampled in 14 populations of Alnus cremastogyne in Sichuan province. [The column heading should show sample size or number of trees sampled. "Size" is too vague]. 

Line 95. Then, the PCR products were submitted for capillary electrophoresis on an ABI … and the fragment lengths were determined … State College, PA, USA. [If post-PCR multiplexing was used, it should be mentioned here]. 

Line 100. Table 2. Characterization of 25 simple sequence repeat loci in Alnus cremastogynebased on 175 trees representing 14 populations in Sichuan province. 

Line 101. Many words in this line should start with lower case letters. 

Line 104. delete "the" 

Line 110. If you calculate G'st, the formula should include it rather than Fst 

Line 115. was performed using Polysat based on 

Line 119. For cluster values of K = 1 to K = 10, 

Line 121. burn-in period was set to 100,000; MCMC after the burn-in period was w35 5o 200,000, and for each K value the calculation was repeated 10 times. The method of Evanno [28] was used … 

Line 127. elevation difference, the Mantel test was performed by NTSYS-pc2.10s 

Line 131. Amplification of 175 A. cremastogyne individuals representing 14 populations using 25 SSR primer pairs generated a total of 271 alleles (Table 2), for a mean of 10.84 alleles at each locus. 

Line 133. 0.934 and from 0.288 to 0.990, with means of 0.665 and 0.739, respectively. 

Line 136. delete "detected" 

Line 137. deled "of the loci" 

Line 138. excess in the A. cremastogyne populations, 

Line 143. Ho ranged from 0.706 to 0.800, and Hs ranged from 0.593 to 0.682. Values for Ht were identical to those for Hs. 

Line 154. Table 3. Genetic diversity in 175 Alnus cremastogyne trees representing 14 populations in Sichuan province as detected by allele sizes at 25 simple sequence repeat loci. 

Line 158. ranged from 0.106 (Act1) to -0.035 (Act15) over the 25 loci, with a mean of 0.021. 

Line 161. populations. Therefore, genetic variation within populations was the main source of variation. The average 

Line 165. This is supported by the observation that Ho was higher than Hs or Ht in all 14 populations. In other words, there was an excess of heterozygotes. 

Line 168. delete "with differences among pairwise populations observed," 

Line 170. Among pairwise comparisons, the value 

Line 174. delete "and it was difficultu for populations to differentiate due to genetic drift". I think the authors mean to say gene flow through pollen or seeds. Genetic drift is random change in allele frequency due to small sample size, and is not the case in this study. 

Line 176. Table S1. Values of Fst (below the diagonal) and Nm (above the diagonal) for pairwise comparison of A. cremastogyne populations estimated using 25 simple sequence repeat loci. 

Line 177. Delete. It is not needed if the Table caption is expanded. 

Line 181. calculated from F-statistics … genetic variation in A. cremastogyne mainly occurs within populations. 

Line 188. Prefecture, and all others 

Line 189. genetic variation in 

Line 192. geographic differentiation 

Line 194. that at K=3, … indicating that the 14 A. cremastogyne populations could be divided … 

Line 196. is shown (Figure 4). At K=3 

Line 200. In the first cluster, 

Line 204. that of the UPGMA clustering analysis 

Line 205. distance, also showed that the 14 A. cremastogyne populations could be divided into three main groups 

Line 211. individuals. 

Line 214. among populations and geographical distance as well as elevation difference using the Mantel test. The correlation coefficient between genetic distance and geographical distance [r = 0.187 (p=0.0760] indicated that 

Line 218. replace "altitude" with "elevation" 

Line 226. In early SSR-related research 

Line 228. Mandak et al [34] were located in the Iberian Peninsula 

Line 236. delete "At the chromosome level" 

Line 237. 56 chromosomes and the basic number of the genus Alnus is x = 14 according to 

Line 240. delete "of two" 

Line 249. geographical variation in 

Line 257. in this study 

Line 258. that are perennial. 

Line 260 species with a narrow distribution 

Line 262. was relatively low (Fst) 

Line 264. similar to delete "located" 

Line 265. Wright [42] considered values of FST in the ranges of 0 – 0.05, … or > 0.25 to represent low … 

Line 268. indicates that gene flow 

Line 272. populations, and those located in the mountainous areas around the Sichuan basin … 

Line 281. change "altitude" to "elevation". 

Line 282. change "altitude distance" to "elevation difference" 

Line 285. The Liangshan and Ganzi areas in this study belong to the southeastern border of the Qinghai-Tibetan Plateau. 

Line 290. the Cretaceous period (?) 

Line 300. delete "the distribution of" 

Line 313. A. cremastogyne and its population genetic diversity. 

Line 314. exhibited the greatest genetic diversity 

Line 315. considered priorities 

Line 316. considered to meet plant breeding needs. 

Line 318. as a pioneer tree species has received little attention. 

Line 321. establishment of its successors. Together with its preference for light … it led to a fragmented distribution … 

Line 327. first study to utilize SSR markers to analyze 

Line 328. delete "A. cremastogyne" 

Line 330. There was little genetic variation among the A. cremastogyne populations; the majority of the variation was within populations. 

Line 336. genetic variation 

Lines 339, 341. estimated using 25 SSR loci.

 References. Most words in the article titles should start with lower case letters. 

Response 1: We have made careful revisions in response to all your suggestions. 

Point 2: The analysis was based on the assumption that Alnus cremastogyne is a tetraploid species, yet this is not mentioned until very late in the text (line 237). It should be mentioned at the beginning so that readers understand the choice of software and the methods of analysis. 

Response 2: In this study, we could not confirm that Alnus cremastogyne is a tetraploid species, as shown in the Discussion section, but the SSR genotype data obtained here revealed the characteristics of a tetraploid, so we have added “For the polyploid characteristics” to “2.3 Data analysis.” 

Point 3: The taxonomic authority should be given only once (Alnus cremastogyne Burk. and Alnus Mill.). After that, only the scientific name is needed. 

Response 3: Because Alnus Mill is commonly known as alder, “alder” was used after the first initial use of “Alnus Mill.” However, there is no well-known common name for Alnus cremastogyne Burk., so we relied on “A. cremastogyne” for subsequent mentions. 

Point 4: The "number of effective alleles" is a term that may be unfamiliar to many readers. For this reason, the term should be defined, how it is calculated should be stated, and how it is interpreted in the context of this study should be mentioned. 

From http://www.uwyo.edu/dbmcd/molmark/waap.html 

AE is the effective number of alleles (at the level of the OTUs we are examining). Verbally, this measure is the number of equally frequent alleles it would take to achieve a given level of gene diversity. That is, it allows us to compare populations where the number and distributions of alleles differ drastically. The formula is: 

where Dj is the gene diversity of the jth of r loci. Note that we calculate the OTU-level AE by averaging over the AE calculated locus-by-locus rather than by calculating a mean gene diversity and then calculating AE from that. The graph below shows why: AE is a nonlinear function of the gene diversity (Hexp), which brings into play Jensen’s inequality [the expectation of a function ≠ the function of the expectations for nonlinear curves; see Ruel and Ayres (1999)]. Here, because the curve is concave up, the AE we compute will be greater than if we calculated it from the overall gene diversity. 

Response 4: We are truly grateful to you for providing this link and information concerning AE, which has helped us to gain more insight into the term and into genetic diversity. However, the “number of effective alleles” is a universal term that describes population genetic diversity, and is not specific to this study. We also refer to some papers (“Genetic Diversity of Walnut (Juglans Regia L.) in the Eastern Italian Alps,” “Genetic Variation in Quercus acutissima Carruth., in Traditional Japanese Rural Forests and Agricultural Landscapes, Revealed by Chloroplast Microsatellite Markers,” “Population Genetic Diversity of Quercus ilex subsp. ballota (Desf.) Samp. Reveals Divergence in Recent and Evolutionary Migration Rates in the Spanish Dehesas,” “Genetic Diversity and Population Genetic Structure of Erythrophleum fordii Oliv., an Endangered Rosewood Species in South China,” “Genetic Diversity and Structure of Natural Quercus variabilis Population in China as Revealed by Microsatellites Markers,” and others) in Forests, and none of them define the term “AE”. For this reason, we believe it is not necessary to define this term in our manuscript. 

Point 5: Line 24. What is "mean genetic variation parameter"? 

Response 5: Response 5: We checked this paragraph carefully, and we have replaced “mean genetic variation parameter” with “the average genetic differentiation” . 

Point 6: Line 36. contains non-leguminous 

Response 6: We believe that the meaning of “Contain” is to include something, so “is” is more appropriate here. 

Point 7: Line 113. Nei's genetic distance between populations (is there a symbol?) 

Response 7: We consulted some papers and previous studies, but we could not find a symbol for Nei’s genetic distance, so we regret we have no way to abbreviate it. 

Point 8: Line 148. was relatively high and populations F (Mianning) and AA (Meigu) had the greatest diversity … In contrast, populations … 

Response 8: The other reviewer also commended that all values are relatively high, so we changed the wording of this sentence to “populations F (Mianning) and AA (Meigu) had the greatest diversity, followed by population B (Luding). By comparison, populations R (Yanting), I (Pingwu), and K (Jiange) showed a slightly lower level of genetic diversity.” 

Point 9: Line 155. Readers should be referred to the map or Table 1 for the locations of the populations. ALL terms should be defined in the footnote. 

Response 9: For easier reading, we rearranged the order of ID in Table 1 in step with order in Table 3, and we also added population ID in Figure 1. All terms are defined in the footnote in Table 3. 

Point 10: Line 212. Figure 3 is not needed. It could be deleted or moved to a supplementary figure. 

Response 10: We reclassified Figure 3 as a supplementary figure (Figure S1). 

Point 11: Lines 239-248. I assume that these are observations of chromosomes during meiosis? 

Response 11: About this point, we checked this reference, and these are observations of chromosomes during mitosis. And karyotyping is to observe chromosome number and morphology during mitosis. 

Point 12: Lines 250-261. Please note that comparisons are possible if the SAME SSR markers are used in the various studies. Otherwise the comparisons become more general. 

Response 12: We regret that SSR markers differ across various studies, so our comparisons must be somewhat general. 

Point 13: Lines 265-270. Also consider/discuss the origins of the populations and the reforestation practices used in the area. 

Response 13: About this point, we mentioned the origins of the populations of A. cremastogyne, human disturbance and the reforestation practices in the area investigated in the following two paragraphs. We mentioned the origins of the populations of A. cremastogyne, human disturbance, and the reforestation practices in the investigated area in the following two paragraphs. 

Point 14: Line 279. seeds have small wings (?) 

Response 14: About this point, the noun “nutlet” may be more appropriate than “seed.” However, in practice, alder nutlets are usually regarded as seeds. So this sentence has been changed to “its nutlets (seeds) have wings and are very lightweight.” The nutlet wings are as shown in the following figures. 

Point 15: Line 307. Birch (Betula) and hazelnut (Corylus) are also in the family Betulaceae, and some species are native to the areas investigated. It would be appropriate to mention them. 

Response 15: Birch (Betula) and hazelnut (Corylus) are located in the areas investigated, especially hazelnuts, including C. yunnanensis, C. heterophylla Fisch. var. sutchuenensis, C. mandshurica, which are also studied in our laboratory. C. yunnanensis is mainly distributed in Liangshan Prefecture, and C. heterophylla Fisch. var. sutchuenensis is mainly distributed in Aba Prefecture, and they have niche overlap in some places, such as Luding. We collected 145 samples from 8 populations, and we are dissecting their genetic diversity and genetic differentiation with 9 SSR markers. We will prepare anther manuscript about hazelnuts in the future. This current study mainly concerns genetic diversity and population structure of A. cremastogyne, which is why it does not mention hazelnut or birch populations. 

Once again, thank you very much for your comments and suggestions. 

Sincerely yours, 

Hong-ying Guo